# Plakophilin-2 Haploinsufficiency Causes Calcium Handling Deficits and Modulates the Cardiac Response Towards Stress

**DOI:** 10.3390/ijms20174076

**Published:** 2019-08-21

**Authors:** Chantal J.M. van Opbergen, Maartje Noorman, Anna Pfenniger, Jaël S. Copier, Sarah H. Vermij, Zhen Li, Roel van der Nagel, Mingliang Zhang, Jacques M.T. de Bakker, Aaron M. Glass, Peter J. Mohler, Steven M. Taffet, Marc A. Vos, Harold V.M. van Rijen, Mario Delmar, Toon A.B. van Veen

**Affiliations:** 1Department of Medical Physiology, Division of Heart & Lungs, University Medical Center Utrecht, Yalelaan 50, Utrecht 3584CM, The Netherlands; 2Division of Cardiology, NYU School of Medicine, New York, NY 10016, USA; 3Institute of Biochemistry and Molecular Medicine, University of Bern, Bern 3012, Switzerland; 4Department of Medical Biology, Academic Medical Center Amsterdam, Amsterdam 1105AZ, The Netherlands; 5Department of Microbiology and Immunology, SUNY Upstate Medical University, Syracuse, NY 13210, USA; 6Dorothy M. Davis Heart and Lung Research Institute, The Ohio State University College of Medicine and Wexner Medical Center, Columbus, OH 43210, USA; 7Departments of Physiology & Cell Biology and Internal Medicine, Division of Cardiovascular Medicine, The Ohio State University College of Medicine Wexner Medical Center, Columbus, OH 43210, USA

**Keywords:** arrhythmogenic cardiomyopathy, plakophilin-2, second hit, calcium handling, fibrosis, exercise, cardiac pressure overload, inflammation

## Abstract

Human variants in plakophilin-2 (PKP2) associate with most cases of familial arrhythmogenic cardiomyopathy (ACM). Recent studies show that PKP2 not only maintains intercellular coupling, but also regulates transcription of genes involved in Ca^2+^ cycling and cardiac rhythm. ACM penetrance is low and it remains uncertain, which genetic and environmental modifiers are crucial for developing the cardiomyopathy. In this study, heterozygous PKP2 knock-out mice (PKP2-Hz) were used to investigate the influence of exercise, pressure overload, and inflammation on a PKP2-related disease progression. In PKP2-Hz mice, protein levels of Ca^2+^-handling proteins were reduced compared to wildtype (WT). PKP2-Hz hearts exposed to voluntary exercise training showed right ventricular lateral connexin43 expression, right ventricular conduction slowing, and a higher susceptibility towards arrhythmias. Pressure overload increased levels of fibrosis in PKP2-Hz hearts, without affecting the susceptibility towards arrhythmias. Experimental autoimmune myocarditis caused more severe subepicardial fibrosis, cell death, and inflammatory infiltrates in PKP2-Hz hearts than in WT. To conclude, PKP2 haploinsufficiency in the murine heart modulates the cardiac response to environmental modifiers via different mechanisms. Exercise upon PKP2 deficiency induces a pro-arrhythmic cardiac remodeling, likely based on impaired Ca^2+^ cycling and electrical conduction, versus structural remodeling. Pathophysiological stimuli mainly exaggerate the fibrotic and inflammatory response.

## 1. Introduction

Arrhythmogenic right ventricular cardiomyopathy (ACM) is an inherited heart disease characterized by fibrous or fibrofatty infiltration of the cardiac muscle, ventricular arrhythmias, and increased propensity for sudden death. Sudden cardiac arrest most often occurs in early adulthood during the subclinical (or “concealed”) phase of the disease, when overt cardiomyopathy is not yet detectable by imaging, and is the first disease manifestation in a high proportion of probands [1,2,3]. ACM associates primarily with variations in genes coding for desmosomal proteins. One of the most commonly mutated genes in ACM is *PKP2*, which encodes the desmosomal protein plakophilin-2 (PKP2) [4,5]. Genetic variants are transmitted autosomal dominantly and prevalence of symptom onset varies [6]. Penetrance of the disease is low and it remains unknown which genetic and environmental modifiers are required for development of the cardiomyopathy. Understanding cardiac remodeling upon environmental stress factors is therefore paramount to understand the mechanisms underlying sudden death and deterioration of cardiac performance.

PKP2 was first described as a component of the desmosome, a cellular structure involved in cell–cell adhesion [7]. Desmosomal dysfunction results in intercellular gap widening and apoptosis of the cardiomyocytes [8,9]. As a result, cardiac progenitor cells and cardiac myocytes may differentiate into fatty or fibrous tissue, proceeding into fibrous and fatty infiltrates that disrupt normal cardiac electrical impulse propagation [10]. Recent studies show that the intercalated disk (ID; including PKP2) not only maintains intercellular coupling, but also modulates transcription pathways fundamental for intracellular Ca^2+^ cycling and cardiac rhythm [5,11]. PKP2 variants lead to phenotypes that vary from purely arrhythmogenic to severe mechanical dysfunction and therefore *pkp2* alterations are also linked to inherited cardiac conditions as Brugada syndrome and Catecholaminergic polymorphic ventricular tachycardia (CPVT) [12,13,14]. Total knockout of *Pkp2* in mice is embryonically lethal, although heterozygous PKP2 deletion in mice does not induce clear phenotypical manifestations and mice live through adulthood. Expression levels and localization of desmosomal proteins (other than PKP2), adherens junction protein, and gap junctional protein are not changed in PKP2 haploinsufficient (PKP2-Hz) mice [15]. Ultrastructural analysis however revealed an increased average intercellular spacing and reduced number and length of mechanical junctions in PKP2-Hz mice [16]. In addition, PKP2-Hz mice present a reduced peak sodium current density and a negative shift of *I*_Na_ steady-state inactivation, which remains masked without addition of sodium channel blockers. Provocation with the sodium channel blocker flecainide in PKP2-Hz animals impairs ventricular conduction, triggers ventricular arrhythmias, and predisposes to sudden death [15].

Additional to *PKP2* variants, several additional factors (secondary hits) have been proposed to contribute to ACM development. Participation in endurance exercise is a key risk factor for developing ACM, its progression toward heart failure, and for the occurrence of arrhythmias and sudden death [17,18,19]. In that regard, patients with ACM are often recommended to avoid endurance training [20]. To date, the involvement of cardiac pressure overload in developing ACM is not well studied yet. Left ventricular hypertrophy induced by cardiac pressure overload is a known precursor of heart failure with severe prognosis [21]. Pressure overload in combination with a loss of desmosomal integrity elevates mechanical stress in cardiac tissue, disturbs intracellular homeostasis, and activates stress-related pathways [5]. Inflammation is a common finding in ACM, the inflammatory response can be caused by cell death, viral infection, or be a consequence of defective desmosomes [22,23,24]. Genetic variants underlying ACM can induce immune alterations that make the heart more vulnerable for myocarditis [25,26]. Separate studies have revealed elevated levels of serum inflammatory mediators and myocardial expression of IL-17 and TNF-alpha in patients with ACM [23]. As well, acute myocarditis reflects an active phase of ACM and accelerates ACM [27]. Conversely, the genetic background can influence the susceptibility towards this superimposed myocarditis [27].

Here, we examined the influence of exercise, cardiac pressure overload, and autoimmune myocarditis on the progression of ACM using a mouse model of PKP2 haploinsufficiency. Hearts of PKP2-Hz mice showed a reduced expression of Ca^2+^-handling-related proteins, confirming intracellular Ca^2+^ disturbances as shown upon total loss of PKP2 expression [11]. Reduced expression of PKP2 exaggerated the (subepicardial) fibrotic and inflammatory response towards pathophysiologic stimuli as pressure overload and inflammation. Exercise-induced pro-arrhythmic cardiac remodeling in PKP2-Hz hearts is likely based on impaired Ca^2+^ cycling and a disturbed electrical conduction, instead of structural remodeling. Showing that PKP2 deficiency controls the cardiac response towards environmental modifiers via different mechanisms.

## 2. Results

### 2.1. PKP2 Haploinsufficiency Impairs Expression of Calcium Handling-Related Proteins

In this study we used a heterozygous PKP2 knock-out (PKP2-Hz) mouse model, as described in Cerrone et al. 2012 (Appendix A) [15]. Since not all patients with a mutation in *PKP2* show symptoms of ACM, additional factors likely contribute to disease development. Such factors could be pressure overload, exercise, and inflammation. We evaluated the potential contribution of several factors to pathogenesis in PKP2-Hz mice (Appendix A). The study of Cerrone et al. showed that levels of intercalated disk proteins N-cadherin (Ncad), connexin 43 (Cx43), plakoglobin (PKG), and Na_v_1.5 are normal in PKP2-Hz hearts, although the sodium current density and kinetics are affected [15]. Total loss of PKP2 expression in mice modulated the transcription of genes important for Ca^2+^ handling and cardiac rhythm, inducing cardiac arrhythmogenesis in the concealed stage of the disease [11]. In the present study we first examined the protein levels of Ca^2+^-handling-related proteins via western blot on ventricular lysates of three-month-old wildtype (WT) and PKP2-Hz mice (Figure 1). In PKP2-Hz mice, ankyrinB (AnkB) expression was on average 27% reduced (*p* = 0.08), calsequestrin-2 (Casq2) 38% (1.0 ± 0.06 vs. 0.62 ± 0.03; *p* ≤ 0.05), Ca_v_1.2 22% (1.0 ± 0.068 vs. 0.78 ± 0.04; *p* ≤ 0.05) and SERCA2a 50% (*p* = 0.08; Figure 1). We confirmed these changes over the timespan of our intervention study. In six-month-old PKP2-Hz mice AnkB was 24% reduced (1.0 ± 0.07 vs. 0.76 ± 0.08; *p* ≤ 0.05), SERCA2a 46% (*p* = 0.08) and Casq2 31%(1.0 ± 0.087 vs. 0.69 ± 0.04; *p* ≤ 0.05; Appendix A). Notably, impaired AnkB expression has been recently associated with human ACM [28].

### 2.2. Characterization of PKP2-Hz Hearts Over Time

Since cardiac electrical and structural remodeling progresses with age, we compared three-month- to six-month-old WT and PKP2-Hz murine hearts [29,30]. Histological analysis by hematoxylin and eosin (H&E) and Picrosirius red revealed no structural modifications at both ages and genotypes (Figure 2A,B), which was confirmed by RNA expression analysis of collagen- and inflammation-related genes (Figure 2C). Timp2 expression was increased (0.83 ± 0.18 vs. 1.98 ± 0.40, *p* ≤ 0.05) in three-month-old PKP2-Hz mice, but this effect did not persist over time (Figure 2C). The heart weight/body weight (HW/BW) index decreased over time, but did not differ between WT and PKP2-Hz mice (Figure 2D and Appendix A). Electrophysiological parameters were unchanged and arrhythmia incidence was rare in six-month-old WT hearts (10%; Figure 2E and Appendix A). Western blot and immunohistochemistry revealed no difference in expression and localization of Ncad and Cx43 (Figure 2F,G). PKG expression however was 49% lower in six-month-old PKP2-Hz hearts compared to WT (1.0 ± 0.13 vs. 0.51 ± 0.03, *p* ≤ 0.05; Figure 2C).

### 2.3. Pro-Arrhythmic Cardiac Remodeling in PKP2-Hz Mice Exposed to Exercise

As endurance training decreases cardiac function and provokes ventricular arrhythmias [17,18], PKP2-Hz mice were subjected to one-month voluntary running on a treadmill. PKP2-Hz mice exposed to exercise training did not show increased levels of fibrosis (Figure 3A, Appendix A), signs of hypertrophy (Figure 3B,C, Appendix A), affected cardiac contractility (Figure 3F), or changes in electrocardiogram (ECG) parameters (Appendix A), compared to controls. The arrhythmia incidence was clearly higher in trained PKP2-Hz mice compared to trained WT animals (PKP2-Hz vs. WT; 40% vs. 0%; Figure 3H). Impaired intracellular Ca^2+^ dynamics (as shown in the three- and six-month-old mice) in combination with endurance exercise possibly increases the susceptibility towards arrhythmias in PKP2-Hz hearts [11].

To confirm our previous findings in three- and six-month-old mice, we examined Ca^2+^ handling-related protein levels in ventricular lysates of WT and PKP2-Hz hearts exposed to exercise training. In PKP2-Hz hearts the expression of AnkB, Casq2, and Ca_v_1.2 was significantly lower than in WT hearts (AnkB 49%: 1.0 ± 0.14 vs. 0.51 ± 0.066, *p* ≤ 0.05; Ca_v_1.2 37%: 1.0 ± 0.08 vs. 0.63 ± 0.097, *p* ≤ 0.05; and Casq2 49%: 1.0 ± 0.08 vs. 0.51 ± 0.02, *p* ≤ 0.05; Figure 3D). We then investigated proteins involved in myocardial cell–cell coupling. Cx43 expression was consistent between WT and PKP2-Hz hearts exposed to exercise training, whereas levels of Ncad and PKG seemed lower in PKP2-Hz hearts than in WT (Ncad; 1.0 ± 0.15 vs. 0.67 ± 0.04, *p* ≤ 0.05, and PKG; 1.0 ± 0.17 vs. 0.58 ± 0.08, *p* = 0.08; Appendix A). Impaired Ncad and PKG expression implicates disturbed mechanical coupling between cardiomyocytes in PKP2-Hz hearts that could affect electrical signaling between cells.

To study the electrical cell–cell coupling in trained mice we analyzed Cx43 localization by immunofluorescence and measured the ventricular conduction velocity by epicardial mapping. PKP2-Hz mice exposed to voluntary exercise presented right ventricular lateral Cx43 expression (Figure 3E) and longitudinal right ventricular conduction slowing, compared to WT mice (left ventricle—LV: 72.10 ± 9.97 vs. 67.43 ± 1.94, ns.; and right ventricle—RV: 59.98 ± 2.58 vs. 52.78 ± 6.11, *p* < 0.05; Figure 3G). The transversal ventricular conduction velocity did not differ between groups (Figure 3G).

### 2.4. Cardiac Pressure Overload is Pro-Fibrotic in the PKP2 Happloinsufficient Heart

Trans aortic constriction (TAC) surgery induces extensive electrical and structural remodeling of the heart and increases arrhythmia vulnerability [31]. To study if pressure overload in combination with loss of PKP2 expression excaggarates the disease progression, we performed TAC surgery on PKP2-Hz and WT mice. Eight weeks after TAC surgery, in both groups hypertrophy was observed (HW/BW WT-Sham: 5.95 ± 0.26 mg/g; WT-TAC: 8.60 ± 1.99 mg/g, *p* ≤ 0.05; PKP2-Hz Sham: 5.84 ± 0.92 mg/g; and 8.44 ± 1.85 mg/g; *p* ≤ 0.01), fractional shortening was reduced (WT-Sham: 48.54% ± 1.2%; WT-TAC: 37.43% ± 3.0%, *p* ≤ 0.0001; PKP2-Hz Sham: 47.3% ± 2.9%; and PKP2-Hz TAC 37.34% ± 5.5%, *p* ≤ 0.001) and QRS duration prolonged (WT-Sham: 9.18 ± 0.5 ms; WT-TAC: 11.01 ± 0.9 ms, *p* ≤ 0.05; PKP2-Hz Sham: 8.69 ± 0.4 ms, and PKP2-Hz TAC: 11.51 ± 0.6 ms, *p* ≤ 0.05). These aspects of cardiac remodeling however did not differ between WT and PKP2-Hz mice (Figure 4A–D, Appendix A). Ventricular repolarization (QTc) was similar between groups (Figure 4D). The susceptibility to sustained ventricular arrhythmias mildly increased due to TAC surgery, with only slightly higher levels in PKP2-Hz TAC than in WT TAC mice (WT TAC vs. PKP2-Hz TAC; 10% vs. 20%; Figure 4E).

To further investigate the determinants of arrhythmogenicity in these mice, we performed tissue analyses for expression of cell-coupling proteins and interstitial fibrosis. Immunolabelling revealed no difference in expression and localization of NCad and Cx43 (Figure 4F). Sirius red staining for collagen abundance presented significant higher levels of interstitial (but not subepicardial) fibrosis in PKP2-Hz TAC mice compared with sham-treated WT and PKP2-Hz mice (WT-Sham: 1.64% ± 0.1%; WT-TAC 2.83% ± 0.95%; PKP2-Hz Sham: 1.86% ± 0.29%; and PKP2-Hz TAC: 5.0% ± 0.86%, *p* ≤ 0.05 vs. WT-Sham and PKP2-Hz Sham; Figure 4G,H). To explore if the Wnt signalling pathway was involved in this pro-fibrotic response, we examined protein levels of β-catenin in ventricular lysates of all groups. Levels of β-catenin were significantly higher in ventricular lysates of TAC mice and this difference was more pronounced in PKP2-Hz mice compared to WT (Appendix A).

### 2.5. PKP2 Haploinsufficiency Alters the Cardiac Fibrotic and Inflammatory Response to Auto-Immune Myocarditis

Inflammation, in the form of a viral infection or acute inflammatory response, has been suggested to trigger arrhtyhmogenic cardiomyopathy [23]. To investigate whether inflammation indeed affects disease progression upon loss of PKP2 expression, we induced experimental autoimmune myocarditis (EAM) in PKP2-Hz and WT mice. Cardiac structural remodeling was studied after three weeks (innate immune response) and six weeks (chronic immune response) of EAM onset. Picrosirius red staining revealed no difference in the collagen content of whole-heart sections, neither after three nor six weeks of EAM (Figure 5A,B). Three weeks after EAM, PKP2-Hz hearts tended to have more subepicardial fibrosis, while after six weeks, this difference was significant (WT six weeks: 6.24% ± 1.6%; and PKP2-Hz six weeks: 14.16% ± 3.3%, *p* ≤ 0.05; Figure 5A,B). By determining the ratio of damaged area to total tissue, more subepicardial damage was found in PKP2-Hz compared to WT mice, three and six weeks after EAM induction, and most pronounced after six weeks (three weeks WT: 5.04% ± 1.4%; three weeks PKP2-Hz: 10.17% ± 2.0%; six weeks WT: 6.02% ± 1.1%; six weeks PKP2-Hz 20.16% ± 3.8%, *p* ≤ 0.05 vs. three weeks PKP2-hz and *p* ≤ 0.001 vs. six weeks WT; WT untreated: 1.34% ± 0.5%, *p* ≤ 0.05 vs. six weeks WT; and PKP2-Hz untreated: 2.76% ± 1.5%, *p* ≤ 0.05 vs. six weeks PKP2-Hz; Figure 5C,D). These observations corresponded to the findings in Cx43 and vimentin-myosin staining of three and six week EAM hearts. PKP2-Hz mice showed a loss of Cx43 expression and larger non-myocyte areas than WTs, three and six weeks after EAM induction (three weeks WT: 0.08% ± 0.02%; PKP2-Hz: 0.18% ± 0.02%, *p* ≤ 0.01, six weeks WT: 0.12% ± 0.03%; and PKP2-Hz: 0.23% ± 0.05%, *p* ≤ 0.05; Figure 5E,F). On top of that, in the 0.2 mm subepicardial layer, PKP2-Hz mice showed a higher neutrophil-positive area than WT mice three weeks pro-EAM onset (Figure 5G). No neutrophils were present in the six-weeks group (data not shown) because the initial immune response most likely had subsided at that timepoint. This subepicardial response of the PKP2-Hz heart towards inflammation did not cause alterations in cardiac contractility or ECG parameters (Appendix A).

## 3. Discussion

ACM is seen as a disease of the intercalated disc (ID) and human variants in PKP2 associate with most familial ACM cases. PKP2 is mainly a component of the desmosome, a cellular structure involved in cell–cell adhesion [5,11,32]. Besides PKP2, genetic variants in other desmosomal genes (Desmocollin2; *DSC2*, Desmoglein2; *DSG2*, and Plakoglobin; *PKG*), have been linked to ACM as well [33]. Since the environmental and genetic modifiers of ACM remain unknown, we examined the effect of exercise, cardiac pressure overload, and inflammation on the progression of a PKP2-related cardiomyopathy. We show that PKP2 deficiency modulated the cardiac response to environmental modifiers via different mechanisms. Exercise induced a pro-arrhythmic cardiac remodeling, likely based on a combination of impaired Ca^2+^ cycling and electrical conduction. Upon pathophysiological stimuli, PKP2 haploinsufficiency mainly exaggerated the fibrotic and inflammatory response.

First of all, we found that PKP2 haploinsufficient mice displayed deficits in Ca^2+^-handling-related proteins, most critically of Ca_V_1.2, SERCA2a, AnkB, and Casq2. This correlates with findings in PKP2 conditional global knock-out (PKP2cKO) mice, in which the expression of proteins involved in Ca^2+^_i_ cycling is markedly reduced [11]. Altered Ca^2+^_i_ transients and Cx43 disfunction potentially causes the ventricular arrhythmias in these mice [11,34]. In addition, PKP2cKO mice present biventricular mechanical dysfunction and fibrosis formation, both starting in the right ventricle [11]. Spontaneous ventricular arrhythmias were observed before the onset of LV dysfunction, followed by end-stage failure and sudden death [11]. In contrast, PKP2-Hz hearts did not display arrhythmogenic events, suggesting that a 50% reduction of PKP2 did not suffice to provoke spontaneous ectopic activity in the heart. It is tempting to speculate that a combination of factors was needed to induce pathological (electrical) remodeling upon PKP2 deficiency. In humans, PKP2 variants are transmitted in an autosomal-dominant manner and mutation carriers can proceed throughout life without developing any signs of cardiomyopathy, similar as observed in the PKP2-Hz mouse model [6,15,35].

In mice, heterozygous loss of PKP2 expression associates with ultrastructural defects, sodium current deficits and hampered sodium channel inactivation kinetics [15,16]. However, under baseline conditions PKP2-Hz mice appear farily nomal and lack an overt disease phenotype [15]. In line with the literature, three- or six-month-old PKP2-Hz mice displayed no structural or electrical abnormalities. This might be explained by the relatively young age of the mice. We acknowledged that six months of age is still mild ageing in mice and increasing the age towards 18–22 months would probably enhance fibrosis formation and arrhythmia incidence [36]. Senescent mice become arrhythmogenic by increased interstitial fibrosis levels and redistributed Cx43 [36]. However, whether extensive aging will discriminate between WT and PKP2-Hz mice remains to be determined.

In line with the literature we observed that PKP2-Hz hearts of mice exposed to one month of voluntary running displayed a higher vulnerability towards sustained ventricular arrhythmias, although lacking any signs of fibrosis [9,12]. While in humans ACM penetrance is low, exercise also greatly increases the risk for developing the cardiomyopathy and for its progression toward failure [17,19]. Life-threatening ventricular arrhythmias or sudden death often occur in the concealed phase of the disease, prior to overt structural damage [37]. The underlying mechanisms responsible for these arrhythmias remain under study but a disturbed electrical coupling could be an important factor [15]. Acute adrenergic stimulation of PKP2 null mice associates with malignant ventricular arrhythmias and sudden death, mimicking exercise during the concealed phase of the disease [11]. In accordance, exercise training in PKP2-R735 mutant mice appeared crucial for developing the ACM phenotype [18]. In addition, we show that a 50% reduction of PKP2 expression was not associated with exercise-induced contractile impairment, in contrast to the study of Cruz et al. [18]. Important to note is the difference in the PKP2 model (PKP2-Hz vs. AAV-induced PKP2-R735 mice), the extent of exercise training (one month of running vs. eight weeks of swimming), and the contractile parameters examined (left ventricular fractional shortening (LVFS) vs. right ventricular ejection fraction (RVEF)) [18]. For future studies, we recommend to explore RV contractility in PKP2-Hz mice exposed to exercise training. Sudden death upon exercise was not observed, and PKP2-Hz mice and their hearts lack any signs of hypertrophy. However, we could not control the intensity of training in our voluntary exercise experiments. Chronic isoproterenol infusion in PKP2-Hz mice would allow studying the effect of adrenergic stimulation on disease development in a more controlled experimental setting [38].

Exercise training of PKP2-Hz mice associates with RV conduction slowing, potentially caused by a combination of impaired mechanical coupling, reduced peak sodium current and redistributed Cx43 [15,18,36]. The absence of structural remodeling in our PKP2-Hz mice suggests that exercise-induced pro-arrhythmic remodeling in PKP2-Hz hearts is caused by an impaired cardiac electrical conduction. The reduced expression of Ca^2+^-handling-related proteins in trained PKP2-Hz hearts is likely not pro-arrhythmic in itself. Though it is tempting to speculate that a combination of impaired intracellular Ca^2+^ cycling and impaired intercellular electrical coupling serves as a pro-arrhythmic substrate in exercise-exposed PKP2-Hz hearts. This needs to be confirmed by in vivo recordings of ventricular arrhythmias in PKP2-Hz mice during exercise, for example by telemetry.

Cardiac pressure overload is a well-known contributor to electrical and structural remodeling of the heart and a risk factor for cardiac arrhythmias [39,40]. In line with the literature we observed that TAC surgery induced hypertrophy, impaired cardiac contractility, and increased susceptibility for arrhythmias; however, PKP2-Hz mice presented similar effects as WT mice [21,39]. TAC surgery increased collagen abundance predominantly in PKP2-Hz hearts. This implicates that PKP2 is involved in regulating the cardiac fibrotic response, as PKP2 haploinsufficiency might lead to activation of pro-fibrotic pathways, inflammation, and apoptosis [40,41]. Enhanced ß-catenin levels in PKP2-Hz TAC mice might confirm this hypothesis, since a loss of desmosomal integrity alters the Wnt-signalling pathway, a well-known contributor to adipo-and fibrogenensis and ACM pathology [42]. Imbalance of the intracellular Ca^2+^ homeostasis and hypertrophic signals possibly increase calcineurin activation in the heart and subsequent inflammatory-driven fibrosis in PKP2-Hz TAC mice [43]. Additional studies are needed to confirm these hypotheses.

In this study we exposed PKP2-Hz mice to an experimental auto-immune myocarditis and studied the acute and innate immune response of the heart. Whole heart collagen levels were not affected by the intervention, although PKP2-Hz mice developed severe levels of subepicardial fibrosis and myocardial damage six weeks after EAM onset. Inflammation was a common finding in ACM. The inflammatory response could be caused by cell death, viral infection, or be a consequence of defective desmosomes [44,45]. Endomyocardial biopsies of ACM patients show features of myocarditis-like inflammation or fibrofatty replacement, possibly reflecting different stages of the disease [27,46]. Acute myocarditis reflects an active phase of ACM and is a superimposed phenomenon during the natural history of the disease [27]. Cardiac fibrofatty replacement in desmosomal mutation carriers is mainly located in the posterolateral wall of the heart, confirming the observations we saw in PKP2-Hz mice exposed to EAM [47]. The molecular mechanism underlying this epicardial response is still under debate. Paracrine signaling between PKP2-positive and PKP2-negative cells in the epicardial region of PKP2-Hz mice is a likely causative factor, activating a TGF-β1/p38 MAPK-dependent fibrotic gene expression [48]. Inflammatory infiltrates were already present at three weeks post-EAM onset in PKP2-Hz mice, suggesting an inflammation-driven fibrotic response. We concluded that PKP2 deficiency altered the cardiac inflammatory and fibrotic response to EAM. This study fitted the hypothesis that ACM might be triggered by infection, as desmosomal mutations compromised the resilience of the heart to inflammation. Inflammation itself may also be arrhythmogenic as previous data suggest that cytokines can alter sodium current function [49]. Future studies are needed to unravel the exact arrhythmic potential of EAM in the PKP2 haploinsufficient heart.

In conclusion, reduced expression of PKP2 in the murine heart affects Ca^2+^-cycling-related protein levels and modulates the cardiac reponse towards environmental modifiers. Cardiac pressure overload and auto-immune myocarditis drive (subepicardial) pro-fibrotic mechanisms in the PKP2 happloinsufficient heart. Exercise-induced pro-arrhythmic cardiac remodeling in hearts with reduced expression of PKP2 is based upon impaired electrical conduction instead of structural remodeling. Our data presented potential important disease mechanisms in a PKP2-related cardiomyopathy.

## 4. Study Limitations

The data presented in this study suggest that decreased levels of Ca^2+^-handling proteins played a maladaptive role in the development of different cardiac phenotypes upon environmantal modifiers in PKP2-Hz mice. Levels of Ca^2+^-handling-related proteins were not examined in mice exposed to cardiac pressure overload and autoimmune myocarditis, so whether cardiac remodeling and Ca^2+^-handling disturbances were directly linked in these models remains uncertain. In addition, no intracellular Ca^2+^ measurements were executed. More in-depth in vitro studies need to be performed to confirm the definite effect of PKP2 haploinsuffiency on cardiac Ca^2+^ dynamics. Another limitation of this study was the lack of arrhythmia score in PKP2-Hz hearts exposed to autoimmunce myocarditis. Histological examination showed that pathophysiological stimuli mainly exacerbated the fibrotic and inflammatory response, but pro-arrhythmic remodeling in PKP2-Hz hearts exposed to autoimmune myocardits could not be excluded

## 5. Materials and Methods

### 5.1. Mouse Model

Mice heterozygous-null for the PKP2 gene (PKP2-Hz) was generated and genotyped as described previously [7]. Except where noted, experiments were conducted in PKP2-Hz and WT littermate C57BL/6J mice (Charles River, Lyon, France) of both genders. For the EAM studies, mice were backcrossed onto the BALB/c background. Animal experiments were performed according to institutional guidelines, the Dutch Experiments on Animals Act and the New York University and SUNY Upstate Medical University guidelines for animal use and care (2010.II.06.109 (04.10.2010), 2010.II.01.006 (09-02-2011), 2011.II.08.116 (15-10-2011), #177 (01-11-2009). Experiments were approved by the local Animal Experiments Committees and conformed to the Guide for the Care and Use of Laboratory Animals published by the US National Institutes of Health.

### 5.2. Transverse Aortic Constriction Procedure

Wildtype and PKP-Hz 12-week-old mice were TAC or sham operated as explained previously [50]. Briefly, mice were anesthetized by isoflurane (mean 2.5% in oxygen), intubated with a 20 G polyethylene catheter, and ventilated (200 µL, 160 strokes/min) with a rodent ventilator (Minivent, Hugo Sachs Electronics, March-Hugstetten, Germany). The thoracic cavity was accessed through a small incision at the left upper sternal border in the second intercostal space. A 7-0 silk suture was passed around the aorta between the right innominate and left common carotid arteries. Constriction of the transverse aorta was performed by tying against a 27 G needle, which was subsequently removed. Sham-operated mice underwent the same procedure without aortic binding. Eight weeks after surgery echocardiograms and electrocardiograms were recorded, the mice were sacrificed by cervical dislocation and the hearts were removed for histological studies.

### 5.3. Exercise Protocol

Wildtype and PKP2-Hz 12-week-old mice were housed solitary in a cage with a treadmill for one month, where they were exposed to daily voluntary exercise training. After one-month voluntary exercise training, echocardiograms and electrocardiograms were recorded, mice were sacrificed by cervical dislocation and the hearts were removed for histological and electrophysiological studies.

### 5.4. Experimental Autoimmune Myocarditis

Myocarditis was induced by immunization with cardiac α-myosin heavy chain (AnaSpec Inc., Fremont, CA, USA) as explained previously [51]. 100 µg cardiac α-myosin heavy chain was dissolved in 100 µL sterile phosphate buffered saline (PBS) and mixed with 100 µL of complete Freund’s adjuvant (Sigma, F5881, st. Louis, MO, USA). This mixture was vortexed for 1.5 h to create an emulsion of antigen in complete Freund’s adjuvant. On day 0, 200 µL of emulsion was injected subcutaneously at two flanks of wildtype and PKP2-Hz mice. For this study, mice were backcrossed onto the BALB/c background as it is permissive for myocarditis [51]. In addition, the mice were injected intraperitoneally with 500 ng of pertussis toxin (List Biological Laboratories, Campbell, CA, USA) dissolved in PBS. The mice were monitored for at least one week following the injections. A mouse was euthanized if it showed unusually large granulomas, ulcerations, or signs of being moribund. After three or six weeks, echocardiograms and electrocardiograms were recorded, the mice were sacrificed by CO_2_ euthanasia and the hearts were removed for histological studies.

### 5.5. Echocardiography

Transthoracic echocardiography was performed using a Vevo2100 Imaging System (VisualSonics Inc., Toronto, Canada) with a 30 MHz probe. Briefly, after induction of anesthesia in a chamber containing isoflurane 4%–5% in oxygen, the mouse was positioned supinely on a heat pad in order to maintain a body temperature at 37–38 °C and anesthesia was maintained with 1.5% isoflurane in 700 mL O_2_/minute via a nose cone. Recordings were obtained in parasternal long and short axis views. Quantitative measurements were assessed offline using the Vevo2100 analytical software. A B-mode parasternal long axis view was used for left ventricular ejection fraction. Left ventricular fractional shortening was calculated from the parasternal short axis view (M-mode) [52].

### 5.6. Electrocardiograms Recordings

Mice were anesthetized with 1.5% isoflurane in 700 mL O_2_ per minute via a nose cone (following induction in a chamber containing isoflurane 4%–5% in oxygen). The rectal temperature was monitored continuously and maintained at 37–38 °C using a heat pad. Three lead ECG (leads I, II, and III) were recorded from sterile needle electrodes inserted subcutaneously in each forelimb and hindlimb. The signal was then acquired and analyzed using a digital acquisition and analysis system (Power Lab; AD Instruments, Oxford, UK; LabChart 7Pro software version). ECG parameters were quantified after 1–2 min from anesthesia induction, in order to stabilize the trace. The QT interval was defined as the time elapsed from the beginning of the major deflection representing the QRS to the end of the secondary slow deflection, as described by Danik et al. [53]. QT intervals were corrected for RR interval by the equation (QTc = QT/(RR/100)1/2), according to Mitchell et al. [54]. Analysis was performed on lead II electrocardiograms.

### 5.7. Epicardial Activation Mapping in Langendorf-Perfused Hearts

For epicardial activation mapping experiments, mice were anesthetized (4% isoflurane in oxygen) and the hearts were quickly excised, rinsed, and placed on a Langendorf column for retrograde coronary perfusion. Hearts were continuously perfused with a Tyrode solution containing: (in mmol/L): NaCl 116, KCL 5, MgSO_4_ 1.1, NaH_2_PO_4_ 0.35, NaHCO_3_ 27, glucose 10, mannitol 16, and CaCl_2_ 1.8 at 37 °C. Solution was continuously gassed with 95% O_2_ and 5% CO_2_. Electrograms were recorded using a 247-point multiterminal electrode (19 × 13 mm grid, 0.3 mm spacing) placed over both the LV and RV, as described before [36,50]. Recordings of the LV and RV were made during stimulation (2 ms duration, 2× diastolic stimulation threshold) from the center of the grid at a basic cycle length (BCL) of 120 ms. The moment of maximal negative dV/dt in the unipolar electrograms was selected as the time of local activation and determined using customized software [55]. Conduction velocity parallel (CV_L_) and perpendicular to fiber orientation were determined from activation maps generated from BCL-pacing. Activation times of at least four consecutive electrode terminals along lines perpendicular to intersecting isochronal lines were used to calculate CVs.

### 5.8. Western Blot Analysis

Total murine heart lysate was prepared as described previously [56]. Equal amounts of protein (25 μg/lane) of each sample were separated on 7% (for Ca_V_1.2 and AnkyrinB) or 10% SDS-polyacrylamide gels (for the remaining proteins) and transferred by electrophoresis to nitrocellulose membranes (Bio-rad, Veenendaan, Netherlands). Equal protein loading was assessed by Ponceau S staining. After the first and second antibody incubation, immunoreactivity was detected using an ECL chemiluminescence kit (Amersham, Piscataway, NJ, USA). The following antibodies were used for western blot: PKP2 (1:250; Fitzgerald, 10R+P130b), plakoglobin (1:500; Sigma, P8087), Cx43 (1:250; Thermo scientific (Landsmeer, Netherlands), 71-0700), N-cadherin (1:10000; Sigma, C12821), calsequestrin-2 (1:1000, Thermo scientific (Landsmeer, Netherlands), PA1-913), ankyrinB (1:500, custom-made in Peter Mohler lab, city, OH, USA), SERCA2a (1:1000, abcam (Camebridge, UK), Ab2861), Ca_V_1.2 (1:500, alomone (Jerusalem, Israel), ACC-003), and β-catenin (1:10000, BD transduction (Vianen, Netherlands), 610154). Secondary labeling was performed with peroxidase-conjugated secondary antibodies (1:7000, Jackson ImmunoResearch, Uden, Netherlands). Images for each western blot, and the corresponding Ponceau S staining, were imported into ImageJ (2008, NIH, Bethesda, MD, USA). The density of each band in the Western blot was normalized to the corresponding Ponceau S signal (after background subtraction) to correct for loading.

### 5.9. PCR Validation of Gene Expression

Total RNA was extracted from the heart tissue of PKP2-Hz and control mice using an RNeasy Mini Kit (QIAGEN, Hilden, Germany). The complement DNA (cDNA) was generated by reverse transcription (RT)-PCR with SuperScript VILO cDNA Synthesis Kit (Thermo scientific, Landsmeer, Netherlands). For RT-qPCR on ventricular tissue, TaqMan gene expression assays (all from Applied Biosystems by Life Technologies Corp., Carlsbad, CA, USA) were used as described earlier [57]. As an internal control, the geometric mean of the TATA-binding protein (TBP) and glyceraldehyde 3-phosphate dehydrogenase (GAPDH) was used. Relative mRNA expression levels were determined for collagen 1α1 (Col1α1), collagen 1α2 (Col1α2), metalloproteinase 9 (MMP9), metallopeptidase inhibitor 1 (Timp1), nuclear factor kappa-light-chain-enhancer of activated B cells (NF-κß), and interleukin-6 (IL-6). Assay IDs are listed in Appendix A.

### 5.10. Immunofluorescence Labeling

For the detection of protein localization by immunofluorescence, hearts were rapidly frozen in liquid nitrogen and stored at −80 °C. Hearts were sectioned (thickness: 10 μm) parallel to the long axis of the heart. Sections displaying a four-chamber view were used for immunostaining as previously described [56]. Samples were exposed to the primary antibodies Cx43 (1:250, BD transduction, 610062), N-cadherin (1:1000, Sigma, C3678), chicken anti-vimentin (1:400, EMD Millipore Corporation (Burlington, MA, USA), AB5733), and myosin (1:200, Enzo Life Sciences Inc. (Bruxelles, Belgium), ALX-BC-1150-S-L001) for primary labeling. Afterwards, Cx43- and Ncad-labeled sections were incubated with FITC conjugated anti-rabbit whole IgG or Texas Red-conjugated anti-mouse whole IgG. Vimentin- and myosin-labeled sections were incubated with secondary antibodies Alexa Fluor 488 goat anti-chicken IgG (1:500, abcam Inc., ab150169) and Alexa Fluor 568 goat anti-mouse IgG (1:500, Thermo scientific (Landsmeer, Netherlands)), A-11031). After labeling, sections were mounted in Vectashield (Vector Laboratories, Orton Southgate, Peterborough, UK) and visualized by conventional epifluorescence and confocal microscopy (Nikon optiphot-2). To examine the non-myosin area, pictures of 40 × magnification of the most damaged area in five heart areas; apex, left-ventricular base/mid region, and right-ventricular base/mid region, were taken. The perimeter of empty area continuous to the epicardial layer was manually selected (in triplet), excluding myocytes and vessels. The non-myosin area continuous to the epicardium was normalized to the length of the epicardium.

### 5.11. Immunohistochemistry

For detection of inflammatory infiltrates and gap junctions by immunohistochemistry, hearts were fixed with 4% paraformaldehyde in PBS. Hearts were sectioned (thickness: 10 μm) parallel to the long axis of the heart. Labeling was performed using the ImmPRESS anti-rat Ig Reagent kit, peroxidase (Vector Laboratories Inc., MP-7444), and the ImmPACT NovaRED Peroxidase substrate kit (Vector Laboratories Inc., SK-4805) to reduce background staining. Antigen retrieval was performed by incubating the sections in a citrate buffer (Biogenex, HK-080-9K) and heating this for 25–30 min using a steamer (Deni, Keystone Inc., Grand Island, NY, USA). Subsequently the sections were incubated with 0.5% hydrogen peroxide (Fisher Scientific (Landsmeer, Netherlands), H323-500) and incubated in 0.3% normal goat serum (Sigma, G9023) in PBS. After blocking, the samples were incubated with primary antibodies (Cx43, 1:250, Thermo Scientific 71-0700; anti-Ly-6G/Ly-6C, 1:200, Biolegend (San Diego, CA, USA) 108402) diluted in ready-to-use 2.5% normal goat blocking serum. Next, sections were incubated for 30 min with ImmPRESS (anti-rat and anti-rabbit) reagent and incubated in peroxidase substrate working solution. Finally, sections were incubated with Vector Hematoxylin QS nuclear counterstain, Gill’s Formula (Vector Laboratories Inc., H-3404) and mounted with Vectashield (Vector Laboratories, H-1000). Quantification of inflammatory infiltrates was performed by computed analysis via Image J. RGB images were converted to stack images and red, blue, and green channels were separated. Red intensity should be at least 110% of the blue intensity, surface area and percent neutrophil positive area was quantified.

### 5.12. Picrosirius Red Staining

To evaluate the extent of fibrosis, sections were fixed with 4% paraformaldehyde in PBS, stained with Picrosirius red and examined by light microscopy. Briefly, paraffin embedded sections were deparaffinated and rehydrated. After repeated washes, slides were incubated in Picrosirius red working solution pH 2.0 (Picric acid, Sigma-Aldrich (Zwijndrecht, Netherlands), 74069; Sirius Red, Polysciences, Inc. (Hirschberg an der Bergstrasse, Germany) C.I. 35780) for 60 min. Sections were directly transferred to 0.01 M HCl solution and incubated under constant movement. Sections were dehydrated again and a coverslip was placed using Entallan (Merck, Schiphol-Rijk, Netherlands, 107960). Stained sections were scanned at a 40× magnification on a Leica SCN400F Whole Slide Scanner. Quantification of healthy myocardium was performed by computed analysis of the percentage red stained tissue with Image J (NIH), which refers to collagen. Regions of interest were selected manually by lining the tissue of both ventricles and septum. The subepicardial layer was defined as 0.2 mm tissues underneath the epicardial layer. By means of a macro the fraction of red staining to the total area was calculated.

### 5.13. Hematoxylin and Eosin Staining

To evaluate the extent of myocardial damage, sections were fixed with 4% paraformaldehyde in PBS, stained with hematoxylin and eosin according to the manufacturer’s instructions, and examined by light microscopy. Briefly, paraffin embedded sections were incubated at 60 °C for 30 min. Slides were incubated in hematoxylin 2 (Thermo Scientific, 7231) for two minutes and directly transferred to clarifier 2 (Thermo Scientific, 7402) for 15 s. After repeated washes, slides were incubated in a Bluing reagent (Thermo Scientific, 1931423) for one minute. After a wash, slides were exposed to eosin Y (Thermo Scientific, 71204) for 1.5 min. After this step the sections were dehydrated again and a coverslip was placed using Permount Toluene solution. Stained sections were scanned at a 40× magnification on a Leica SCN400F Whole Slide Scanner. Two independent researchers assessed the subepicardial damage in HE-stained hearts, by determining the ratio of the non-myocyte area—continuous of the epicardial layer—to total tissue area. This was done by computed analysis via Image J (NIH) in five windows of 0.59 mm × 0.31 mm over the ventricular wall; apex, left-ventricular base/mid region, and right-ventricular base/mid region. The perimeter of the empty area continuous to the epicardial layer was manually selected (in triplet), excluding myocytes and vessels.

### 5.14. Statistical Analysis

Statistical analysis and drawing of graphs and plots were carried out in GraphPad Prism (version 6 for Mac OS X, GraphPad Software, San Diego, CA, USA) and SigmaStat 3.5 software. Normality and equal variance assumptions were tested with the Kolmogorov-Smirnov and the Levene’s median test, respectively. Differences between two groups were analyzed using the paired two-tailed Student’s *t*-test, comparisons between experimental groups were analyzed by a one-way ANOVA for non-parametric variables with a Tukey’s post-test for intergroup comparisons. All data is presented as mean ± SEM, and *p* < 0.05 was considered significant. * *p* ≤ 0.05, ** *p* ≤ 0.01, *** *p* ≤ 0.001, **** *p* ≤ 0.0001, and *n*.s. *p* > 0.05. *n* denotes the number of mice used per dataset.

## Figures and Tables

**Figure 1 ijms-20-04076-f001:**
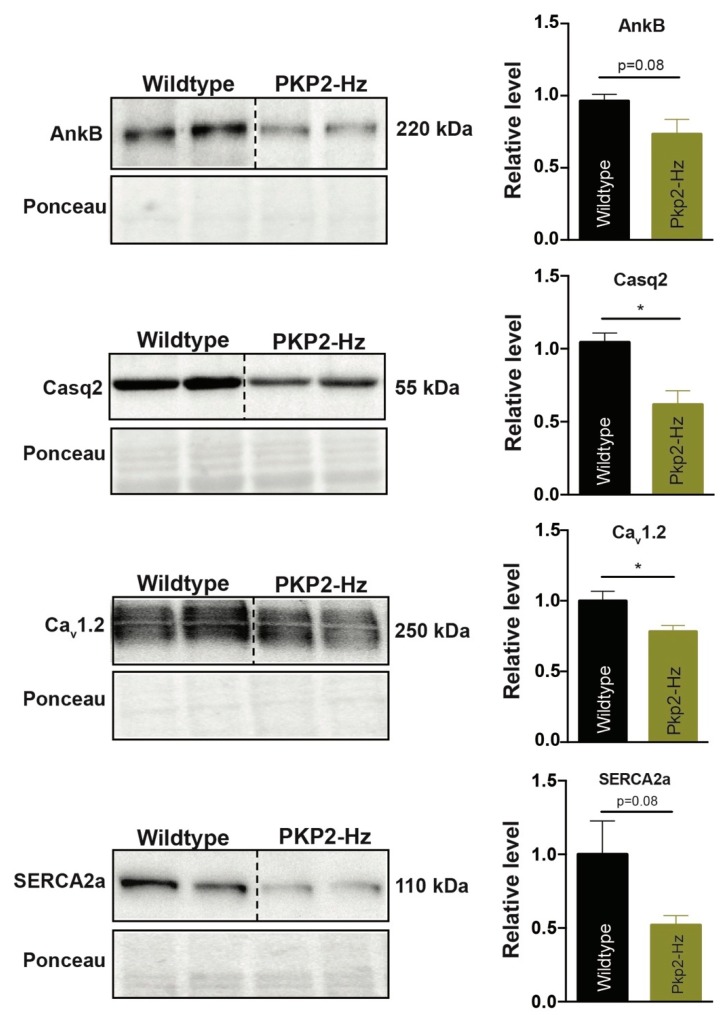
Remodeling of proteins involved in calcium signaling pathways in the PKP2-Hz mouse. Representative western blots (left) and average densitometry (right; *n* = 5 for all groups) of ankyrinB (AnkB), calsequestrin-2 (Casq2), Ca_v_1.2, and SERCA2a, measured from wildtype and PKP2-Hz ventricular lysates of three-month-old mice. Left upper panels represent western blots, bottom panels according to ponceau staining used for quantification (mean ± SEM, * *p* < 0.05).

**Figure 2 ijms-20-04076-f002:**
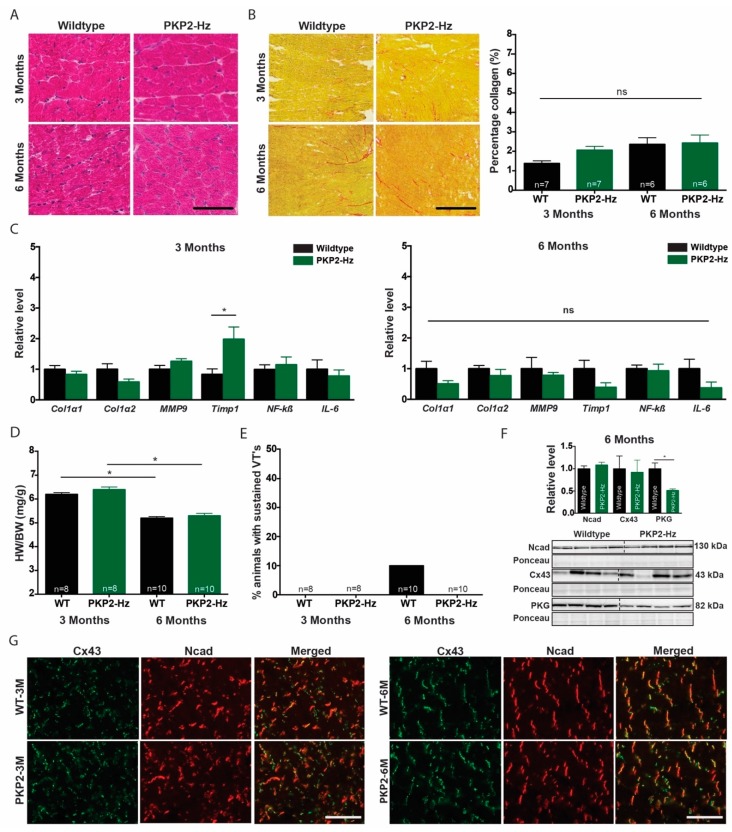
Absence of cardiac remodeling in three and six months old PKP2-Hz mice. (**A**) Hematoxylin and eosin staining of wildtype and PKP2-Hz heart sections, at three and six months of age; scale bar, 100 µm. (**B**) Representative pictures of Picrosirius red staining (left panel) in the inner myocardium of wildtype and PKP2-Hz murine hearts of both ages; scale bar, 500 µm. Quantification of collagen abundance (right panel) in overview slides of whole heart sections (WT 3 months; *n* = 7, PKP2-Hz three months; *n* = 7, WT six months; *n* = 6, PKP2-Hz six months; *n* = 6, mean ± SEM). (**C**) Relative mRNA expression of Col1α1, Col1α2, MM9, Timp1, NF-κß, and IL-6 assessed by RT-qPCR in three (left) and six month old (right), wildtype and PKP2-Hz ventricular tissue (three months; *n* = 6 for all groups, six months; *n* = 4 for all groups, mean ± SEM, * *p* < 0.05). (**D**) Quantification of heart weight (HW) to body weight (BW) ratio (WT three months; *n* = 8, PKP2-Hz three months; *n* = 8, WT six months; *n* = 10, PKP2-Hz six months; *n* = 10, mean ± SEM, * *p* < 0.05). (**E**) Percentage of Langendorf-perfused hearts within each group susceptible for sustained ventricular arrhythmias by epicardial pacing (WT three months; *n* = 8, PKP2-Hz three months; *n* = 8, WT six months; *n* = 10, PKP2-Hz six months; *n* = 10). (**F**) Representative western blots (bottom) and average densitometry of N-cadherin (Ncad), connexin 43 (Cx43), and plakoglobin (PKG; top, WT; *n* = 6, PKP2-Hz *n* = 5 for Cx43 and PKG, WT; *n* = 4, PKP2-Hz *n* = 5 for Ncad), measured from wildtype and PKP2-Hz ventricular lysates, six months of age. Upper panels represent western blots, bottom panels according ponceau staining used for quantification (mean ± SEM, * *p* < 0.05). (**G**) Immunofluorescence staining for Cx43 (green) and Ncad (red) in wildtype (WT) and PKP2-Hz (PKP2) heart sections of three (3M) and six months (6M) of age; Scale bar, 100 µm.

**Figure 3 ijms-20-04076-f003:**
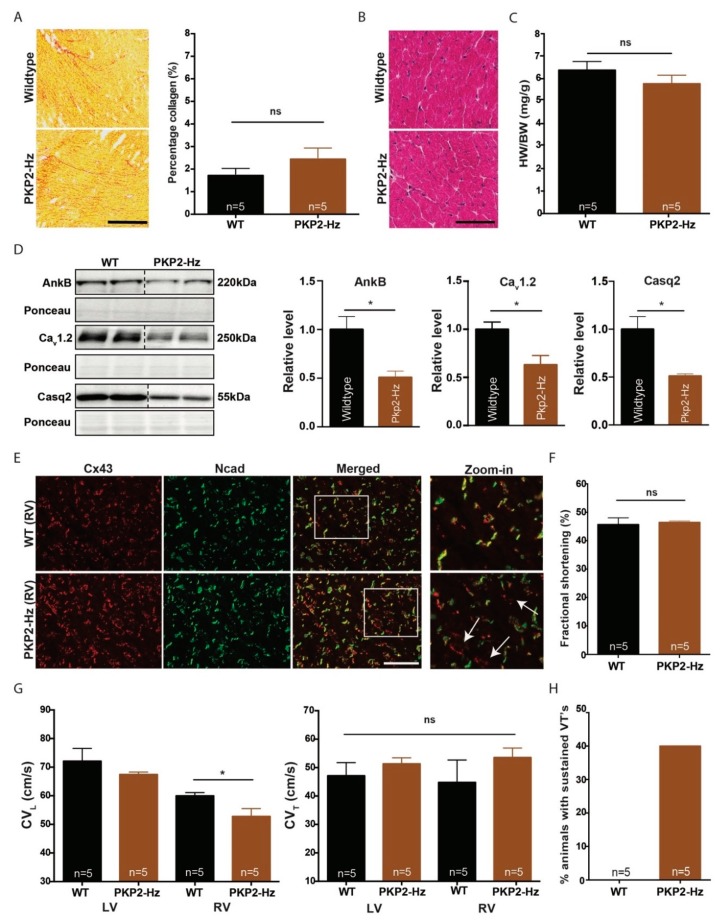
Pro-arrhythmic cardiac remodeling in PKP2-Hz mice exposed to exercise training. (**A**) Representative pictures of Picrosirius red staining (left panel) in the inner myocardium of wildtype and PKP2-Hz murine heart sections, of mice exposed to exercise training; scale bar, 500 µm. Quantification of collagen abundance (right panel) in overview slides of whole heart sections (WT; *n* = 5 and PKP2-Hz; *n* = 5, mean ± SEM). (**B**) Hematoxylin and eosin staining of wildtype and PKP2-Hz heart sections, of mice exposed to exercise; scale bar, 100 µm. (**C**) Quantification of heart weight (HW) to body weight (BW) ratio in both groups (WT; *n* = 5 and PKP2-Hz; *n* = 5, mean ± SEM). (**D**) Representative western blots (left panel) and average densitometry (right panels, *n* = 5 for all groups) of AnkyrinB (ANKB), Ca_v_1.2, and calsequestrin-2 (Casq2) measured from wildtype (WT) and PKP2-Hz ventricular lysates of mice exposed to exercise training. Left upper panels represent western blots, bottom panels according ponceau staining used for quantification (mean ± SEM, * *p* < 0.05). (**E**) Immunofluorescence staining for Cx43 (red) and Ncad (green) in right ventricular heart sections of WT and PKP2-Hz mice; scale bar, 100 µm. Upper right panel shows magnified shots of according merged pictures, indicated by the white boxes. Lateral Cx43 expression is more pronounced in PKP2-Hz hearts exposed to exercise training, indicated by white arrows. (**F**) Left ventricular fractional shortening examined via echocardiography in hearts of wildtype (WT) and PKP2-Hz mice after one-month of training (WT; *n* = 5 and PKP2-Hz; *n* = 5, mean ± SEM). (**G**) Conduction (CV) velocity measured by epicardial mapping on the left ventricle (LV) and right ventricle (RV) in longitudinal (left) and transverse (right) directions in hearts of wildtype (WT) and PKP2-Hz mice (WT; *n* = 5 and PKP2-Hz; *n* = 5, mean ± SEM, * *p* < 0.05). (**H**) Percentage of Langendorf-perfused hearts, within each group, susceptible for sustained ventricular arrhythmias by epicardial pacing (WT; *n* = 5 and PKP2-Hz; *n* = 5).

**Figure 4 ijms-20-04076-f004:**
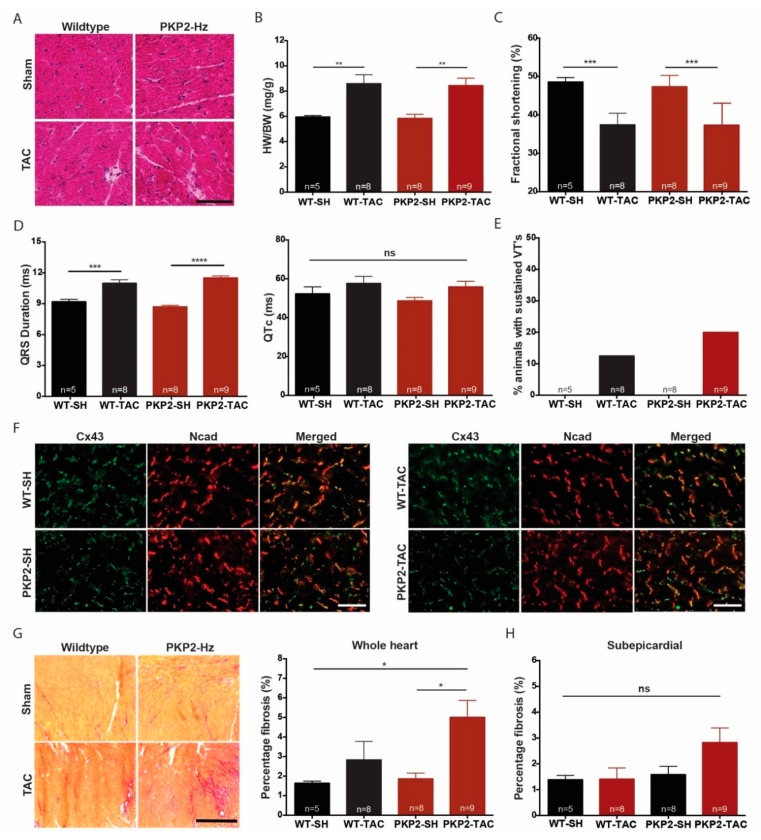
Pro-fibrotic cardiac remodeling in PKP2-Hz trans aortic constriction (TAC) operated mice. (**A**) Hematoxylin and eosin staining of wildtype and PKP2-Hz heart sections of Sham and TAC operated mice; scale bar, 100 µm. (**B**) Quantification of heart weight (HW) to body weight (BW) ratio in all four groups (WT Sham; *n* = 5, PKP2-Hz sham; *n* = 8, WT TAC; *n* = 8, and PKP2-Hz TAC; *n* = 9, mean ± SEM, ** *p* < 0.01). (**C**) Left ventricular fractional shortening examined by echocardiography in wildtype (WT) and PKP2-Hz mice after eight weeks of TAC/Sham surgery (WT Sham; *n* = 5, PKP2-Hz sham; *n* = 8, WT TAC; *n* = 8, and PKP2-Hz TAC; *n* = 9, mean ± SEM, *** *p* < 0.001). (**D**) QRS duration (left) and ventricular repolarization (QT) time corrected for heart rate (QTc; right) examined via electrocardiograms in wildtype (WT) and PKP2-Hz mice after eight weeks of surgery (WT Sham; *n* = 5, PKP2-Hz sham; *n* = 8, WT TAC; *n* = 8, and PKP2-Hz TAC; *n* = 9, mean ± SEM, *** *p* < 0.001). (**E**) Percentage of Langendorf-perfused hearts, within each group, susceptible for sustained ventricular arrhythmias by epicardial pacing (WT Sham; *n* = 5, PKP2-Hz sham; *n* = 8, WT TAC; *n* = 8, and PKP2-Hz TAC; *n* = 9). (**F**) Immunofluorescence staining for Cx43 (green) and Ncad (red) in wildtype and PKP2-Hz (PKP2) heart sections of mice exposed to Sham (SH) or TAC surgery; scale bar, 100 µm. (**G**) Representative pictures of Picrosirius red staining (left panel) in the inner myocardium of all four groups; scale bar, 500 µm. Quantification of collagen abundance (right panel) in overview slides of whole heart sections (WT Sham; *n* = 5, PKP2-Hz sham; *n* = 8, WT TAC; *n* = 8, and PKP2-Hz TAC; *n* = 9, mean ± SEM, * *p* < 0.05). (**H**) Quantification of collagen abundance (right panel) in the 0.2 mm subepicardial layer, examined in overview slides of whole heart sections (WT Sham; *n* = 5, PKP2-Hz sham; *n* = 8, WT TAC; *n* = 8, and PKP2-Hz TAC; *n* = 9, mean ± SEM).

**Figure 5 ijms-20-04076-f005:**
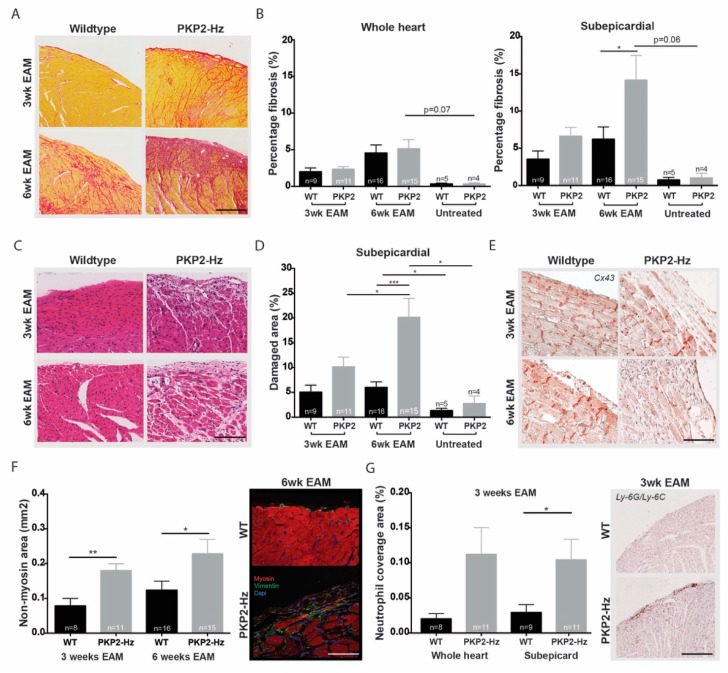
Exaggerated cardiac fibrotic and inflammatory response in PKP2-Hz mice exposed to autoimmune myocarditis. (**A**) Representative pictures of Picrosirius red staining (left panel) of the outer myocardial layer in wildtype and PKP2-Hz murine heart sections, of mice after three weeks of experimental auto-immune myocarditis (3wk EAM) and six weeks of experimental auto-immune myocarditis onset (6wk EAM; scale bar, 750 µm). (**B**) Quantification of collagen abundance in the entire heart (left panel) and 0.2 mm subepicardial region (right panel) of wildtype (WT) and PKP2-Hz (PKP2) mice after three weeks experimental autoimmune myocarditis (EAM) and six weeks EAM onset or without treatment (untreated; WT 3wk EAM; *n* = 9, PKP2 3wk EAM, *n* = 11, WT 6wk EAM; *n* = 16, PKP2 3wk EAM, *n* = 15, WT untreated; *n* = 5, and PKP2 untreated, *n* = 4, mean ± SEM, * *p* < 0.05). (**C**) Hematoxylin and eosin (H&E) staining in wildtype and PKP2-Hz heart sections exposed to EAM; scale bar, 500 µm. **(D**) Quantification of damaged area in the 0.2 mm subepicardial layer of wildtype (WT) and PKP2-Hz (PKP2) mice hearts exposed to EAM or without any treatment, quantification was performed on H&E stained heart sections (WT 3wk EAM; *n* = 9, PKP2 3wk EAM, *n* = 11, WT 6wk EAM; *n* = 16, PKP2 6wk EAM, *n* = 15, WT untreated; *n* = 5, and PKP2 untreated, *n* = 4, mean ± SEM, * *p* < 0.05, *** *p* < 0.001). (**E**) Representative examples of Connexin 43 (Cx43) immunohistochemistry labeling in the subepicardial region of WT and PKP2-Hz heart sections exposed to EAM; scale bar, 100 µm. (**F**) Representative examples (right) of myosin (red) and vimentin (green) immunofluorescence labeling and quantification of non-myosin area (left) in wildtype and PKP2-Hz heart sections of mice exposed to EAM; scale bar, 100 µm. The non-myosin area was examined in the epicardial layer of both ventricles, areas were normalized by the length of the epicardium (WT 3wk EAM; *n* = 8, PKP2 3wk EAM, *n* = 11, WT 6wk EAM; *n* = 16, and PKP2 6wk EAM, *n* = 15, mean ± SEM, * *p* < 0.05, ** *p* < 0.01). (**G**) Quantification of Ly6C/6G (neutrophil) coverage area (left) in whole heart and 0.2 mm subepicardial regions of wildtype and PKP2-Hz mice after three weeks EAM onset (WT whole heart; *n* = 8, PKP2 whole heart, *n* = 11, WT subepicard; *n* = 9, and PKP2 subepicard, *n* = 11, mean ± SEM, * *p* < 0.05). Representative example (right) of the neutrophil positive area in both groups; scale bar, 500 µm.

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
