# Peer review of "Plakophilin-2 Haploinsufficiency Causes Calcium Handling Deficits and Modulates the Cardiac Response Towards Stress"

_ijms, 2019, doi:10.3390/ijms20174076_

Round 1

Reviewer 1 Report

The manuscript by van Opbergen et al. is a very interesting and well written/comprehensive paper. Given the complexity involved, the authors have produced many positive and welcome outcomes.

I have no hesitation in recommending it to be accepted for publication. However, I have just one criticism to this work: For WB, I would avoid the Ponceau S staining as loading control in your WB and, mostly, for the quantification of each band. In my opinion this is an inappropriate method, being an approach much more qualitative than quantitative. I sincerely recommend the authors to use a house keeping control (GAPDH, Actin…etc) employing specific antibodies for loading control and then for densitometric quantification. I am confident that the experiments/controls I suggest will not time-consuming for the authors, indeed, it would improve the reliability and the quality of this already excellent manuscript.

Reviewer 2 Report

Present article showed excellent laboratory data by using PKP2 k/o mouse.  

1. Significance and association with other desmosome genetic variants and ACM may also briefly commented in discussion.

2. I recommend to comment possible association between PKP2 variants and Brugada syndtome.

3. Kind of inflammatory cell appeared in mouses may be shown in more detail.

4. Why experimental autoimmune myocarditis you showed was more severe in aubepicardial region?.

5. Limitation of present study should be shown in more detail.

6. Revierwer recommended to introduce clinical and pathological studies to detect sudden unexpected death with concealed or early clinical ACM. Recent study that try to detect trivial Inflammatory foci as possible result of genetic variant could cause ACM may also be referenced

Author Response

Presented article showed excellent laboratory data by using PKP2 k/o mouse. 

We wish to thank the reviewer for her/his kind comments. We are encouraged by the fact that the reviewer found our laboratory data of excellence. 

Significance and association with other desmosome genetic variants and ACM may also briefly commented in discussion.

Thank you for this comment and specification. We followed the advice and introduced in the discussion that other desmosomal genetic variants are linked to ACM as well (page 12, lines 309-310).

I recommend to comment a possible association between PKP2 variants and Brugada syndrome.

We agree that PKP2 is more than just an ACM susceptibility gene and mutations can lead to phenotypes that vary from purely arrhythmogenic to severe mechanical dysfunction. PKP2 deficiency can therefore indeed lead to other inherited cardiac conditions, such as Brugada syndrome or CPVT (Cerrone et al., 2014, Tester et al., 2019). We followed the recommendation and included the link between PKP2 variants and other disease phenotypes (including Brugada syndrome) in the introduction section on page 2, lines 69-71.

Tester DJ, Ackerman JP, Giudicessi JR, Ackerman NC, Cerrone M, Delmar M, et al. Plakophilin-2 Truncation Variants in Patients Clinically Diagnosed With Catecholaminergic Polymorphic Ventricular Tachycardia and Decedents With Exercise-Associated Autopsy Negative Sudden Unexplained Death in the Young, JACC Clin. Electrophysiol., 2019;5: p120–7. Cerrone M, Lin X, Zhang M, Agullo-Pascual E, Pfenniger A, Chkourko Gusky H, et al. Missense mutations in plakophilin-2 cause sodium current deficit and associate with a Brugada syndrome phenotype, Circulation, 2014;129:1092–103.

Kind of inflammatory cell appeared in mouses may be shown in more detail.

The reviewer’s suggestion is appreciated. Inflammatory cells in PKP2-Hz mice exposed to autoimmune myocarditis is examined with an antibody against lymphocyte antigen6G/6C (Ly-6G/Ly-6C), a marker commonly used to detect neutrophils (Lee et al., 2013, Navarini et al., 2009). As described in the Materials and Methods (page 14&15, lines 453-464), peroxide labeling on whole heart sections was used to detect Ly-6G/Ly-6C positive cells. This approach is very useful for quantitative examination of inflammatory cells, but less appropriate for detailed analysis of inflammatory cell characteristics. High-magnification imaging would not reveal more details on the kind of inflammatory cells and for Figure 5 our goal was to show inflammatory infiltrates in overlapping regions of collagen abundance and tissue damage. We would respectfully submit that stating the type of inflammatory cells, Ly-6G-Ly-6C/neutrophils, is sufficient to appreciate the inflammatory/fibrotic process in PKP2-Hz mice. We added in Figure 5G the labelling which was used to emphasize the type of inflammatory cells.

Lee PY, Wang J, Parisini E, Dascher CC, Nigrovic PA, Ly6 family proteins in neutrophil biology, Journal of Leukocyte Biology, 2013, 94(4), p.585-594 Navarini AA, Lang KS,Verschoor A, Recher M, Zinkernagel AS, Nizet V, et al., Innate immune-induced depletion of bone marrow neutrophils aggravates systemic bacterial infections, Proc Natl Acad Sci U S A, 2009, 106(17): 7107–7112.

Why experimental autoimmune myocarditis you showed was more severe in subepicardial region?

We very much appreciate this question. We like to point out that the experimental autoimmune myocarditis approach itself was not directed to the subepicardial region of the heart and control mice lack a specific subepicardial response. Mice are injected with a cardiac-specific immunization mixture subcutaneously, to induce myocarditis throughout the entire heart. Interestingly, PKP2-Hz mice respond to the autoimmune myocarditis by developing subepicardial fibrosis, inflammation and loss of tissue integrity. Cardiac fibrofatty replacement in human desmosomal mutation carriers is also mainly located in the posterolateral wall of the heart, though the exact mechanism behind this fibrotic infiltration of (sub)epicaridal origin is still unknown (Sepehrkhouy et al., 2017). As speculated in the discussion, paracrine signaling between PKP2-positive and PKP2-negative cells is a likely causative factor, though no definite anwers can be given at this moment (Dubash et al., 2016).

Sepehrkhouy S, Gho JMIH, van Es R, Harakalova M, de Jonge N, Dooijes D, et al. Distinct fibrosis pattern in desmosomal and phospholamban mutation carriers in hereditary cardiomyopathies, Heart Rhythm, 2017; 14: p 1024–32. Dubash AD, Kam CY, Aguado BA, Patel DM, Delmar M, Shea LD, et al. Plakophilin-2 loss promotes TGF-β1/p38 MAPK-dependent fibrotic gene expression in cardiomyocytes, Cell Biol., 2016;212: p425–38.

Limitation of present study should be shown in more detail.

We thank the reviewer for this remark. We have added a paragraph explaining the limitations of this study in more detail, page 14, lines 414-425.

‘’4. Study limitations

The data presented in this study suggest that decreased levels of Ca2+-handling proteins play a maladaptive role in the development of different cardiac phenotypes upon environmantal modifiers in PKP2-Hz mice. Levels of Ca2+-handling-related proteins have not been examined in mice exposed to cardiac pressure overload and autoimmune myocarditis, so whether cardiac remodeling and Ca2+-handling disturbances are directly linked in these models remains uncertain. In addition, no intracellular Ca2+measurements have been excecuted. More in-depth in vitro studies need to be performed to confirm the definite effect of PKP2 haploinsuffiency on cardiac Ca2+dynamics. Another limitation of this study is the lack of arrhythmia score in PKP2-Hz hearts exposed to autoimmunce myocarditis. Histological examination showed that pathophysiological stimuli mainly exacerbate the fibrotic and inflammatory response, but pro-arrhythmic remodeling in PKP2-Hz hearts exposed to autoimmune myocardits cannot be excluded. ‘’

6.Reviewer recommended to introduce clinical and pathological studies to detect sudden unexpected death with concealed or early clinical ACM. Recent study that tries to detect trivial Inflammatory foci as possible result of genetic variant could cause ACM, may also be referenced.

We are not quite sure if we understand this reviewer’s remark correctly. To emphasize the urgency of understanding sudden death mechanisms in concealed-stage ACM we followed the recommendation and referred in the introduction to two cases of sudden cardiac death in young adults, with and without a known secondary hit (exercise) as initiator (Müssigbrodt et al. 2018, Lin Y et al. 2018). Patients with ACM are often recommended to avoid endurance training, as it is a key risk factor for ventricular arrhythmias and sudden death (James et al. 2013). Moreover, endurance training in heterozygous plakoglobin-deficient mice did accelerate the development of right ventricular dysfunction, in absence of any histological abnormalities (Kirchhof et al. 2006). As suggested in our manuscript, cardiac Ca2+handling disturbances are not yet confirmed as prime cause of sudden death in early clinical ACM in patients yet. Although, isoproterenol-induced sudden death in concealed ACM has been observed in cardiomyocyte-specific conditional PKP2 knock-out mice (Cerrone et al. 2017). Important to note is that these mice present severe right-ventricular Ca2+disregulation preceding structural changes (Kim et al. 2019). In addition, abnormal right-ventricular deformation and reduced contractility has been reported in subclinical ACM desmosomal mutation carriers, suggesting right-ventricularCa2+deficitsalready occur in pre-clinical ACM stages (Mast et al. 2016).

The reviewer suggested to refer to a study that  describes trivial inflammatory foci as cause of ACM, without indications of studies the reviewer refers to. We appreciate that genetic variants underlying ACM can indeed induce immune alterations that make the heart more vulnerable for myocarditis (Asimaki et al. 2011). As well, acute myocarditis reflects an active phase of ACM and accelerates ACM (Lopez-Ayala et al. 2015). To emphasize the role of inflammatory foci in ACM disease progression, we refer in the introduction to a pathological studypresenting that scattered foci of lymphocytes and myocardial death were observed in almost 70% of ACM autopsy cases (Basso et al. 1996).

James CA, Bhonsale A, Tichnell C, Murray B, Russell SD, Tandri H, et al. Exercise increases age-related penetrance and arrhythmic risk in arrhythmogenic right ventricular dysplasia/cardiomyopathy-associated desmosomal mutation carriers, J Am Coll Cardiol., 2013; 62: p1290-1297. Kirchhof P, Fabritz L, Zwiener M, Witt H, Schafers M, Zellerhof M, et al., Age- and training-dependent development of arrhythmogenic right ventricular cardiomyopathy in heterozygous plakoglobin-deficient mice, Circulation, 2006, 114(17), p1799-1806. Cerrone M, Montnach J, Lin X, Zhao Y-T, Zhang M, Agullo-Pascual E, et al. Plakophilin-2 is required for transcription of genes that control calcium cycling and cardiac rhythm, Nat. Commun., 2017; 8: p106. Kim J-C, Pérez-Hernández Duran M, Alvarado FJ, Maurya SR, Montnach J, Yin Y, et al. Disruption of Ca2+i Homeostasis and Cx43 Hemichannel Function in the Right Ventricle Precedes Overt Arrhythmogenic Cardiomyopathy in PKP2-Deficient Mice, Circulation.doi: 10.1161/CIRCULATIONAHA.119.039710. Mast TP, Teske AJ, Walmskey J, van der Heijden JF, van ES R, Prinzen FW, et al., Right Ventricular Imaging and Computer Simulation for Electromechanical Substrate Characterization in Arrhythmogenic Right Ventricular Cardiomyopathy, J Am Coll Cardiol., 2016, 68(20); p.2185-2197. Asimaki A, Tandri H, Duffy ER, Winterfield JR, Mackey-Bojack S, Picken MM, et al. Altered desmosomal proteins in granulomatous myocarditis and potential pathogenic links to arrhythmogenic right ventricular cardiomyopathy, Circ. Arrhythm. Electrophysiol., 2011,; 4: p743–52. Lopez-Ayala JM, Pastor-Quirante F, Gonzalez-Carrillo J, Lopez-Cuenca D, Sanchez-Munoz JJ, Oliva-Sandoval MJ, et al. Genetics of myocarditis in arrhythmogenic right ventricular dysplasia. Heart Rhythm, 2015;12:766–73. Basso C, Thiene G, Corrado D, Angelini A, Nava A, Valente M, Arrhythmogenic Right Ventricular Cardiomyopathy: Dysplasia, Dystrophy, or Myocarditis? Cicrulation, 1996, 94; p983-991.

Round 2

Reviewer 1 Report

The referee is satisfied with th authors correction and explications, thus, I strongly recommend the publication of this paper